# *EbARC1*, an E3 Ubiquitin Ligase Gene in *Erigeron breviscapus*, Confers Self-Incompatibility in Transgenic *Arabidopsis thaliana*

**DOI:** 10.3390/ijms21041458

**Published:** 2020-02-20

**Authors:** Mo Chen, Wei Fan, Bing Hao, Wei Zhang, Mi Yan, Yan Zhao, Yanli Liang, Guanze Liu, Yingchun Lu, Guanghui Zhang, Zheng Zhao, Yanru Hu, Shengchao Yang

**Affiliations:** 1State Key Laboratory of Conservation and Utilization of Bio-Resources in Yunnan, The Key Laboratory of Medicinal Plant Biology of Yunnan Province, Yunnan Agricultural University, Kunming 650201, China; 18275621127@163.com; 2National and Local Joint Engineering Research Center on Germplasm Innovation and Utilization of Chinese Medicinal Materials in Southwest China, Yunnan Agricultural University, Kunming 650201, China; fanwei1128@aliyun.com (W.F.); bing.hao@hotmail.com (B.H.); 18468181512@163.com (M.Y.); zhaoyankm@126.com (Y.Z.); liangyanlimt@sina.com (Y.L.); guanzeliu@163.com (G.L.); lyc13888813931@163.com (Y.L.); zgh73107310@163.com (G.Z.); 3CAS Key Laboratory of Tropical Plant Resources and Sustainable Use, Xishuangbanna Tropical Botanical Garden, Chinese Academy of Sciences, Kunming 650223, China; 4College of Life Science and Technology, Honghe University, Mengzi 661100, China; zw_biology2@126.com; 5College of Agriculture and Life Sciences, Kunming University, Kunming 650214, China; zhaozheng2118@163.com

**Keywords:** *Erigeron breviscapus*, protein interaction, ARC1, Exo70A1, Self-incompatibility

## Abstract

*Erigeron breviscapus* (Vant.) Hand.-Mazz. is a famous traditional Chinese medicine that has positive effects on the treatment of cardiovascular and cerebrovascular diseases. With the increase of market demand (RMB 500 million per year) and the sharp decrease of wild resources, it is an urgent task to cultivate high-quality and high-yield varieties of *E. breviscapus*. However, it is difficult to obtain homozygous lines in breeding due to the self-incompatibility (SI) of *E. breviscapus*. Here, we first proved that *E. breviscapus* has sporophyte SI (SSI) characteristics. Characterization of the ARC1 gene in *E. breviscapus* showed that *EbARC1* is a constitutive expression gene located in the nucleus. Overexpression of *EbARC1* in *Arabidopsis thaliana* L. (Col-0) could cause transformation of transgenic lines from self-compatibility (SC) into SI. Yeast two-hybrid (Y2H) and bimolecular fluorescence complementation (BiFC) assays indicated that EbARC1 and EbExo70A1 interact with each other in the nucleus, and the EbARC1-ubox domain and EbExo70A1-N are the key interaction regions, suggesting that EbARC1 may ubiquitinate EbExo70A to regulate SI response. This study of the SSI mechanism in *E. breviscapus* has laid the foundation for further understanding SSI in Asteraceae and breeding *E. breviscapus* varieties.

## 1. Introduction

Sexual reproduction of flowering plants is a complex biological process that involves recognition between pollen and stigma. In order to ensure biodiversity, several effective mechanisms in plants have evolved to avoid inbreeding and promote heterosexuality, such as herkogamy, dichogamy, and self-incompatibility (SI) [1,2,3,4,5]. Herkogamy is the separation of the pistil and stamen in space, while dichogamy is the staggering of the maturation time of the pistil and stamen in time, thus effectively reducing the probability of self-pollination [6]. Unlike herkogamy and dichogamy, SI is a process in which the specific recognition of stigma and pollen rejects self-pollen fertilization [6]. According to the genetic control pattern, SI has been widely accepted as either sporophyte SI (SSI) or gametophyte SI (GSI) [7]. Presently, the GSI mechanism has been well illustrated in Solanaceae, Rosaceae, and Papaveraceae. However, the SSI mechanism in other plants is still poorly understood, except in *Brassica*.

In *Brassica*, SSI is controlled by multiple alleles in the S-locus, including S-locus cysteine-rich protein/S-locus protein 11 (SCR/SP11) and S-locus receptor kinase (SRK), which act as ligands and receptors, respectively [8,9,10]. Firstly, SCR/SP11 recognizes SRK and phosphorylates SRK. Phosphorylated SRK further interacts with ARC1 (E3 ubiquitin ligase) and phosphorylates ARC1 [11,12,13,14,15]. Finally, ARC1 degrades compatibility factors, including Exo70A1, Glyoxalase I (GLO1), and Phospholipase D α1 (PLDα1), by ubiquitination, thus regulating the SI response [16,17,18]. Inhibition of *ARC1* expression in *Brassica napus* L. and *Arabidopsis lyrata* L. could partially break SSI, indicating that ARC1 is relatively conserved in the positive regulation of SSI response [19,20]. However, whether such assumed conservation of ARC1 can also be applied to other families of plants remains to be studied.

Besides *Brassica*, SSI also exists in 73% of Asteraceae plants [21,22]. However, the molecular mechanism of SSI in Asteraceae is not clear. In order to identify the female and male determinants in the S-locus of Asteraceae, a large number of previous studies focused on *Senecio squalidus* L. and found that genes encoding stigma S-associated protein (SSP), Senecio S-receptor-like kinases (SSRLKs), and stigma-specific peroxidase (SSP), which are expressed in the stigma of mature flowers, might be the key candidate genes in the S-locus that mediates the SSI response [23,24,25,26]. However, the specific function of these genes has not been revealed. *Erigeron breviscapus* (Vant.) Hand.-Mazz. is a perennial herbaceous plant of *Erigeron* in Compositae, which is mainly distributed in the mountainous areas of Southwest China. The rosette leaves grow at the basal part of the stem and survive at the flowering stage. The capitulum, which consists of peripheral female ligulate flowers and central hermaphroditic tubular flowers, grows alone or branches at the top of the stem. In the wild, *E. breviscapus* is mainly pollinated by insect vectors, and the life history is R-strategy, which prevails with quantity [27]. Up to now, *E. breviscapus* has been used as a Chinese traditional medicinal plant for more than 1000 years. Breviscapine, one of the most valuable compounds in *E. breviscapus*, has a significant effect on the treatment of cardiovascular and cerebrovascular diseases, with a commercial value of RMB 500 million annually [28]. Our previous studies showed that SSI is a characteristic of *E. breviscapus*. Further transcriptomic analysis showed that the *SRK*, *ARC1*, *CaM*, and *Exo70A1* genes were differentially expressed in self-pollination and cross-pollination, indicating that they may be involved in the SI response of *E. breviscapus* [29]. However, their functions have not yet been characterized.

In this study, we further confirmed that *E. breviscapus* has SSI characteristics. The cloning and functional characterization showed that EbARC1 could interact with the downstream EbExo70A1, and overexpression of *EbARC1* in *Arabidopsis thaliana* L. would lead to the SI response of *transgenic plants*, providing circumstantial evidence that *EbARC1* might mediate the SSI response of *E. breviscapus*.

## 2. Results

### 2.1. E. breviscapus has SSI Characteristics 

Previous studies have shown that both self-compatibility (SC) and SI are present in Asteraceae [22]. To confirm whether *E*. *breviscapus* possesses theSI response, the aniline blue staining method was used to observe pollen tube growth between self-pollination and cross-pollination. The result showed that the pollen tube could grow normally in the style of cross-pollination (Figure 1A), while it was inhibited in the style of self-pollination after germination (Figure 1B). This feature suggests that *E*. *breviscapus* has SSI characteristics.

### 2.2. Molecular Characteristics and Expression Analysis of EbARC1

In order to study the potential role of *ARC1* in the SSI of *E. breviscapus*, the *ARC1* homolog from *E. breviscapus* was cloned and designated as *EbARC1*. The full length of *EbARC1* is 2199 bp, which encodes a protein with 732 amino acid residues. EbARC1 contains the U-box N-terminal domain (UND), U-box domain and ARM repeat domain, which are highly conservative to those in other species (Figure 2A). The phylogenetic analysis showed that EbARC1 is most closely related to AaARC1 of *Artemisia annua* L. (Figure 2B).

The subcellular localization of EbARC1 was determined in a transient expression system in *Nicotiana benthamiana* leaves. The EbARC1::GFP fusion protein was found to be exclusively localized to the nucleus (Figure 3A). The expression profiles of *EbARC1* detected by real-time PCR further indicated that *EbARC1* was expressed in whole plant tissues, with relatively higher expression in cauline leaves and lower expression in buds (Figure 3B), suggesting that *EbARC1* is a constitutive expression gene.

### 2.3. Overexpression of EbARC1 Causes Self-Sterility in A. thaliana

Previous studies have reported that *ARC1* positively regulated the SSI response in *Brassica* and *A. lyrata* [16,18,19,20]. To investigate whether the SSI response in *E. breviscapus* was also regulated by *EbARC1*, we overexpressed *EbARC1* in *A. thaliana* by transforming the recombination plasmid of pocA30-CaMV-*35S*::*EbARC1* (Figure 4A). Two independent transgenic lines *EbARC1-OE-L6* and *EbARC1-OE-L36* were selected to study self-sterility characteristics. We found that the silique length was shorter and the seed number per silique was much less compared with the wild-type plants (Figure 4B–D). In addition, we also observed that stamen height in transgenic lines was lower than stigma compared with wild-type plants (Figure 4E). These results show that EbARC1 mediates the SI response in *A. thaliana*.

### 2.4. Transforming SC into SI through Overexpressing EbARC1 in A. thaliana

To further analyze what causes sterility, we carried out a hybrid test. By analyzing the silique growth and pollen tube staining of two different sets of hybridization (i.e., EbARC1-OE-L36 (♀) × Col-0, and Col-0 (♀) × EbARC1-OE-L36), we found that the pollen and stigma of EbARC1-OE-L36 were normal (Figure 5A–D). Because the stamen of EbARC1-OE-L36 is lower than the stigma, we examined whether the sterility is caused by the pollen not falling on the stigma. The staining analysis of selfing wild type and EbARC1-OE-L36 showed that the stigma of EbARC1-OE-L36 inhibited the growth of pollen tubes, but it was normal in wild type (Figure 5E,F). These results indicate that the sterility was not caused by the activity of the pollen and stigma and the heterotopia of the stigma but was achieved by the SI mediated by the overexpression of *EbARC1*.

### 2.5. EbARC1 Protein Interacts with EbExo70A1

To investigate the potential protein interacting with EbARC1, we fused full-length EbARC1 to the GAL4 DNA-binding domain of the bait vector (BD-EbARC1) and found a positive clone encoding 638 amino acid residues through a yeast two-hybrid (Y2H) screening system. Sequence analysis showed that the positive clone was Exo70A1, which is highly conserved among homologs in *A. annua*, *Cynara cardunculus*, *Lactuca sativa*, and *Helianthus annuus* L. (Appendix A). *EbExo70A1* was expressed in all tissues of *E. breviscapus*, and the highest expression was in buds (Appendix A). Subcellular localization showed that EbExo70A1was also located in the nucleus (Appendix A).

To verify the interaction between EbARC1 and EbExo70A1, we fused the full-length EbExo70A1 to the GAL4 activation domain of the prey vector (AD-EbExo70A1). Y2H indicated that coexpressing four combinations (BD-EbARC1 and AD-EbExo70A1, BD and AD, BD-EbARC1 and AD, BD and AD-EbExo70A1) displayed normal yeast growth in SD-T/L defective medium, whereas yeast only coexpressing BD-EbARC1 and AD-EbExo70A1 could grow in SD-T/L/H/A defective medium, suggesting the interaction between EbARC1 and EbExo70A1 (Figure 6A,B). Moreover, the bimolecular fluorescence complementation (BiFC) assays showed that the coexpressing EbARC1–n-YFP and EbExo70A1–c-YFP in *N. benthamiana* could produce green fluorescence, which further illustrated that the interaction between EbARC1 and EbExo70A1 occurred in the nucleus (Figure 6C).

To analyze the key domains responsible for the interaction between EbARC1 and EbExo70A1, EbARC1 was divided into three parts according to its domain distribution, including BD-EbARC1 (1-588 aa), BD-EbARC1 (1-368 aa), and BD-EbARC1 (1-301 aa), while EbExo70A1 was divided into EbExo70A1-N (1-284 aa) and EbExo70A1-C (285-639 aa). Y2H results showed that EbARC1 (1-368 aa) and EbExo70A1-N (1-284 aa) are the key domains of the two proteins’ interaction (Figure 6A,B).

## 3. Discussion

In Brassicaceae, SSI controlled by multiple alleles in the S-locus (*SCR* and *SRK*) can promote cross-pollination among related species by inhibiting the growth of pollen tubes on the style, ultimately ensuring the biodiversity of plants. However, the mechanism of SSI response in other families is still poorly understood. So far, the S-locus controlling SSI in Asteraceae has not been determined, which seriously hinders the understanding of the SSI mechanism and breeding in Asteraceae. Previous studies have shown that many plant species in Asteraceae are SSI (e.g., *Tolpis coronopifolia (Desf.) Biv.*, *S. squalidus*) [30,31], while a few are SC (*L. sativa* and cultivated sunflower). *E. breviscapus* is a medicinal plant of Asteraceae, which has a significant effect on the treatment of cardiovascular and cerebrovascular diseases [32]. Our previous study found that self-pollination of *E. breviscapus* could not produce seeds [29]. Here, we further found that all the pollen tubes of self-pollinated plants could not grow normally, while most of the pollen tubes of cross-pollinated plants grew normally (Figure 1), confirming that there is SSI in *E. breviscapus*. However, the molecular mechanism controlling SSI in *E. breviscapus* is still unclear. 

Previous studies have found that the structural mutation of S-locus genes (*SCR* and *SRK*) is considered to be an important reason for the transformation from SI into SC in *Arabidopsis* [33,34]. Coexpressing *A. lyrate AlSCR-SRK* in Sha, Kas-2, and C24 ecotypes could transform SC into SI but not for the Col-0 ecotype [35,36]. Further investigation found that this phenomenon is caused by the mutation of ARC1 rather than S-locus genes [20], which was supported by the fact that expression of *SCR-SRK-AlARC1/BnARC1/AhARC1* in Col-0 and Sha could produce significant SI [5,37]. Although *A. lyrate*, *Arabidopsis helleri* L., and *B. napus* belong to Brassicaceae, *Arabidopsis* and *Brassica* differentiated about 20–40 million years ago [38]. Interestingly, *AlARC1/BnARC1/AhARC1* showed the same function in the regulation of SI response [5,14,37], suggesting that the role of *ARC1* derived from different species may be conservative. However, up to now, there is no report on whether high expression of *ARC1* alone in *A. thaliana* (Col-0) can transform SC into SI. In this study, overexpression of *EbARC1* in *A. thaliana* resulted in self-sterility (Figure 4). Further analysis showed that the sterility of the EbARC1-OE-L6/L36 line was caused by the SI response, not by the activity of the pollen and stigma or the heterotopia of the stigma (Figure 5), indicating that EbARC1 was positively involved in SI regulation.

Further analysis showed that EbARC1 interacted with Exo70A1 (Figure 6). The same study in *Brassica* also indicated that ARC1 interacts with Exo70A1, and the ARC1-Ubox domain and the N-terminal of Exo70A1 are the key regions for the interaction [16,39,40], which is consistent with our results (Figure 6). Exo70A1 is a compatibility factor that is required for normal germination and growth of pollen [41]. In *Brassica*, the degradation of Exo70A1 mediated by ARC1 could result in SI [16]. At the same time, the mutation of *Exo70A1* in *A. thaliana* caused dwarf and sterile plants [42]. The prediction showed that the EbExo70A1 protein has multiple potential sites of ubiquitination (Appendix A), suggesting that EbARC1 may ubiquitinate EbExo70A1 through these sites. Meanwhile, the interaction between EbARC1 and EbExo70A1 also implied that there are several crosstalks and coevolutions between Asteraceae and Brassicaceae in SSI response. In addition, several studies have also shown that ARC1 can interact with upstream SRK and M-locus protein kinase (MLPK) [11,14,15]. Recent research on *B. napus* reported that the SI response was partially broken down in the *bnmlpk* mutant due to downregulated expression of *BnSRK* and *BnARC1* [43]. Overexpression of the GATA transcription factor *BnA5.ZML1* could partially breakdown the SI response in *B. napus* by indirectly regulating *SRK* and *ARC1* expression [44]. In the future, using EbARC1 as bait to screen a library will help to identify new SI-related proteins and describe the regulatory network of SI in *E. breviscapus*.

In traditional cross-breeding, inhibiting self-pollination by artificial emasculation or an emasculation agent not only takes time but also increases the breeding cost. In this study, the heterologous expression of *EbARC1* in *A. thaliana* showed that *EbARC1* functions in regulating the SI response by transforming *A. thaliana* SC into SI. Considering that we have preliminarily established the *Agrobacterium*-mediated genetic transformation system of *E. breviscapus*, the site-directed mutation of *ARC1* in *E. breviscapus* through gene editing will help to verify the function of ARC1 and create homozygous parent materials in the future.

## 4. Materials and Methods

### 4.1. Plant Materials and Growth Conditions

The seeds of *Arabidopsis thaliana* L. (Col-0) were preserved in our laboratory, and *EbARC1*-overexpressing *A. thaliana* lines were obtained by *Agrobacterium-*mediated transformation. *A. thaliana* was planted in an artificial greenhouse at 22 °C under a 10 h light/14 h dark condition. The seedlings of *E. breviscapus* were obtained from Longjin Biotech Co., Ltd., Xuanwei, China and then were transplanted into flowerpots (20 cm in length and width, 15 cm in height) for growth in a natural greenhouse. The substrate of all materials was vermiculite: perlite (3:1), and 1/5 Hoagland nutrient solution was poured every 3 days. For *A. thaliana*, phenotype tests and crosses were carried out in three biological repeats (*n* = 20 for each biological repeat). For *E. breviscapus*, three separate biological repeats collected in the same environment during different planting seasons (spring, summer, and autumn) were used to perform the crosses (*n* = 30 for each biological repeat) and subsequent qRT-PCR (*n* = 10 for each biological repeat).

### 4.2. Bioinformation Analysis

The sequences of *EbARC1* and *EbExo70A1* were acquired from an *E. breviscapus* genome database (available online: https://www.herbal-genome.cn). Other species’ protein sequences of ARC1 and Exo70A1 were obtained from the NCBI data bank. Amino acid sequences were analyzed using the Clustal Omega software (available online: https://www.ebi.ac.uk/Tools/msa/clustalo/) and a phylogenetic tree was constructed using the MEGA 6.0 software (available online: https://www.megasoftware.net/mega4/mega.html). The ubiquitin site prediction was conducted on the network UbiSite (available online: http://csb.cse.yzu.edu.tw/UbiSite/predict.php).

### 4.3. Vector Construction and Plasmid Transformation

The coding sequence (CDS) of *EbARC1* was amplified via specific primers that contained SacI/Xbal restriction endonuclease sites for the subcellular localization assay for EbARC1. The amplified CDS of *EbARC1* was inserted into the pocA30-*CaMV-35S*-GFP vector to generate recombinant plasmid. To obtain the overexpressing transgenic plants, the *EbARC1* CDS was inserted into the pocA30-*CaMV-35S* vector via BamHI/Sall restriction endonuclease sites. The reconstructed plasmids were transformed into *Agrobacterium tumefaciens* (strain EHA105) and then transformed into *Col-0* plants by the floral dipping method. Transgenic lines that displayed a 3:1 ratio for hygromycin resistance in the T2 generation were selected for further analysis.

### 4.4. Subcellular Localization Analysis

The recombinant plasmids pocA30-CaMV-35S::*EbARC1*-GFP and control plasmids pocA30-*CaMV-35S*::GFP were transformed into *A. tumefaciens* (strain EHA105). After overnight cell culture at 28 °C, *A. tumefaciens* was harvested by centrifugation and resuspended in infiltration liquid media (0.15 mM acetosyringone, 10 mM MgCl_2_, 10 mM MES-KOH; pH 5.6). Leaf epidermis cells of *N.benthamiana* L. were imaged using a confocal laser scanning microscope (Olympus, Tokyo, Japan). GFP fluorescence was observed at 450 nm excitation and 490 nm emission. The nuclei were also stained with DAPI (4’,6-diamidino-2-phenylindole dihydrochloride) as a control to indicate nucleus localization.

### 4.5. Pollination Assays

For *E. breviscapus,* the stamens of immature flowers were removed at the full flowering stage and crossed with the mature pollen of different plants (*n* = 30 for each biological repeat). For *A. thaliana*, the Col-0 and transgenic lines were emasculated firstly in the stage 12 buds, and the crosses were carried out when the flower was completely open and the stigma was active (*n* = 20 for each biological repeat). After that, the cross buds were covered with plastic film and placed in darkness overnight. Photos of the top 10 siliques per plant were taken with a Nikon digital camera.

### 4.6. Aniline Blue Staining Assay

The aniline blue staining was performed in the stigmas of *E. breviscapus* after self-pollination and cross-pollination for 2 days and in the stigmas of *A. thaliana* after self-pollination and cross-pollination for 2 h, respectively. The collected stigmas were fixed by ethanol and glacial acetic acid (3:1). After that, the stigmas were treated overnight with 1 M NaOH solution and washed several times with potassium metaphosphate solution. Subsequent staining was performed as previously described by Samuel [16]. The stained stigmas were observed using a fluorescence microscope (Olympus-bx63, Olympus, Tokyo, Japan).

### 4.7. RNA Extraction and qRT-PCR

Total RNA was extracted from different tissues of *E. breviscapus* (*n* = 10 for each biological repeat) using the Trizol reagent (Invitrogen, New York, USA). The first-strand cDNA was synthesized from 1.5 μg of DNase-treated RNA in a 20 μL volume system using M-MuLV reverse transcriptase (Thermo, New York, USA) with oligo (dT) 18 primer. qRT-PCR was performed using SYBR Green master mix (Vazyme, Nanjing, China) on a Roche Light-Cycler 480 real-time PCR machine (Roche, Basel, CHE), according to the manufacturer’s instructions. The conditions for PCR amplification were as follows: 94 °C for 5 min; 45 cycles of 94 °C for 10 s, 55 °C for 15 s, and 72 °C for 20 s. The PCR reactions were carried out in three biological and technical repeats, and expression data were normalized with respect to the EbACTIN2 expression level. The expression data were analyzed based on the comparative 2^−ΔΔCT^ method. The primers used in this assay are listed in Appendix A.

### 4.8. Yeast Two-Hybrid Assay

The full-length as well as truncated *EbARC1* were cloned into the bait vector pGBKT7, while the full-length *EbExo70A1* and truncated fragments were cloned into the prey vector pGADT7. Yeast two-hybrid assays were performed according to the protocol described in Clontech’s Matchmaker^TM^ Gold Yeast Two-Hybrid user manual. Primers used for amplifying these CDSs and fragments are listed in Appendix A

### 4.9. BiFC Assay

The full-length *EbExo70A1* CDS was inserted into c-YFP to generate c-YFP–*EbExo70A1*, while *EbARC1* CDSs were inserted into n-YFP to form n-YFP–*EbARC1*. The recombinant plasmids were introduced into *A. tumefaciens* (strain EHA105) and infiltrated *N. benthamiana* young leaves. Infected tissues were observed 48–72 h after infiltration. YFP and DAPI fluorescence were observed at 450 nm excitation and 490 nm emission via a confocal laser scanning microscope (Olympus-FV1000, Olympus, Tokyo, Japan). Primers used for amplifying these full-length and fragmented genes are listed in Appendix A.

## 5. Conclusions

In this study, SSI characteristics in *E. breviscapus* has been uncovered. The functional charaterization of an E3 ubiquitin ligase gene *EbARC1* in *A. thaliana* demonstrated that it cofers SI response of transgenic plants probaly by ubiquiting compatibility factor EbExo70A1, suggesting that the involvement of *EbARC1* in SSI response of *E. breviscapus*.

## Figures and Tables

**Figure 1 ijms-21-01458-f001:**
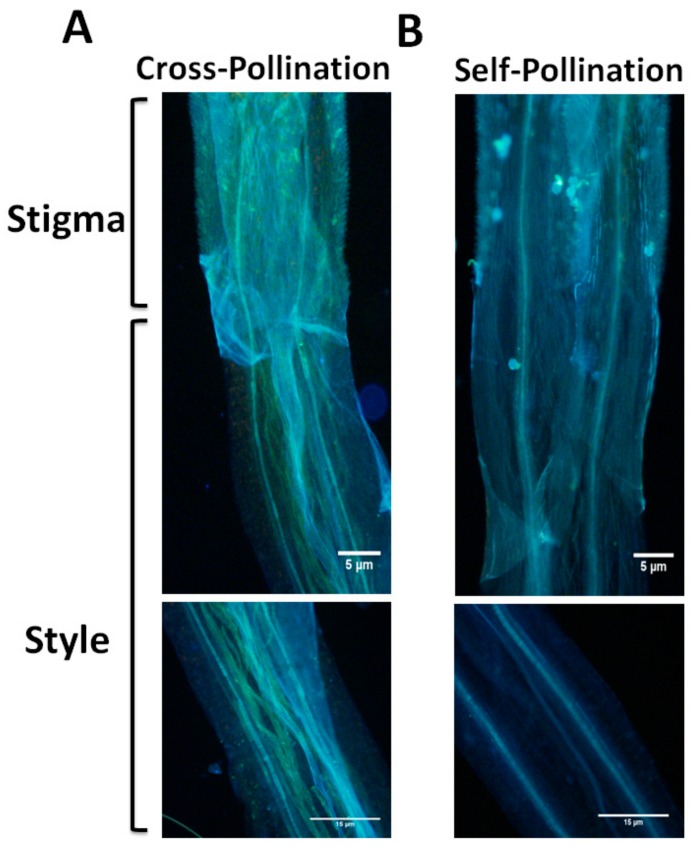
The growth of pollen tubes in the stigma and style of *Erigeron breviscapus* was observed by aniline blue staining 2 days after pollination. (**A**) Cross-pollinated pollen tubes could pass through the stigma. Normally growing pollen tubes can be seen in the style. (**B**) Self-pollination pollen tube growth was suppressed in the stigma, and no pollen tube growth was observed in the style. Three biological repeats were separately carried out in spring, summer, and autumn (*n* = 30).

**Figure 2 ijms-21-01458-f002:**
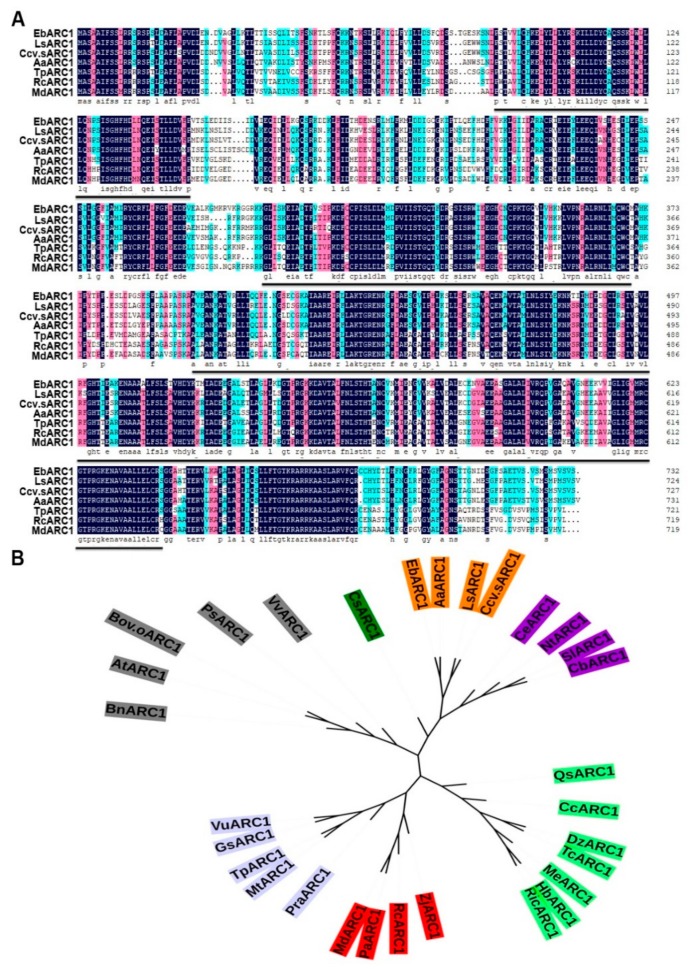
Sequence analysis of EbARC1. (**A**) Alignment of the amino acid sequences of EbARC1 and orthologous proteins from other plant species, including *Lactuca sativa* L. var. ramosa Hort. (XP_023732862.1), *Cynara cardunculus* var. *scolymus* L. (XP_024992831.), *Artemisia annua* L. (PWA78229.1), *Trifolium pretense* L. (PNX95847.19), *Rosa chinensis* Jacq. (XP_024199500.1), and *Malus domestica* L. (XP_008371510.2). Identical residues are shown on a black background. Blue and black lines point to the conserved domains of the U-box and ARM repeat domains, respectively. (**B**) Phylogenetic tree of EbARC1 and ARC1s from other plant species, including *Arabidopsis thaliana* L. (AtARC1, NP_195803.1), *L. sativa* (LsARC1, XP_023732862.1), *A. annua* (AaARC1, PWA78229.1), *C. cardunculus* (Ccv.sARC1, XP_024992831.), *Nicotiana tabacum* L. (NtARC1, NP_001313174.1), *Solanum lycopersicum* L. (SlARC1, XP_004233034.1), *Hevea brasiliensis* L. (HbARC1, XP_021659773.1). *Durio zibethinus* Murr. (DzARC1, XP_022720359.1), *Medicago truncatula* Gaertn. (MtARC1, XP_003593822.2), *Theobroma cacao* L. (TcARC1, EOY27921.1), *Prosopis alba* Grisebach. (PaARC1, XP_028770127.1), *Coffea eugenioides* S. Moore. (CeARC1, XP_027163052.1), *Glycine soja* Sieb. (GsARC1, XP_028188525.1), *Capsicum baccatum* L. (CbARC1, PHT55858.1), *Manihot esculenta* Crantz. (MeARC1, XP_021597342.1), *Vigna unguiculate* L. Walp. (VuARC1, XP_027932349.1), *Ricinus communis* L. (RcARC1, XP_002522266.1), *Trifolium pretense* L. (TpARC1, PNX95847.1), *Quercus suber* L. (QsARC1, XP_023874938.1), *Ziziphus jujube* Mill. (ZjARC1, XP_015875655.1), *Citrus clementina* Hort. (CcARC1, XP_006449554.1), *Rosa chinensis* Jacq. (RcARC1, XP_024199500.1), *Vitis vinifera* L. (VvARC1, RVW19787.1), *Prunus avium* L. (PaARC1, XP_021813189.1), *Camellia sinensis* L. (CsARC1, XP_028072792.1), *Malus domestica* L. (AdARC1, XP_008371510.2), *Brassica napus* L. (BnARC1, XP_022558023.1), *Papaver somniferum* L. (PsARC1, XP_026455241.1), and *Brassica oleracea* var. *oleracea* L. (Bov.oARC 1, XP_013593471.1).

**Figure 3 ijms-21-01458-f003:**
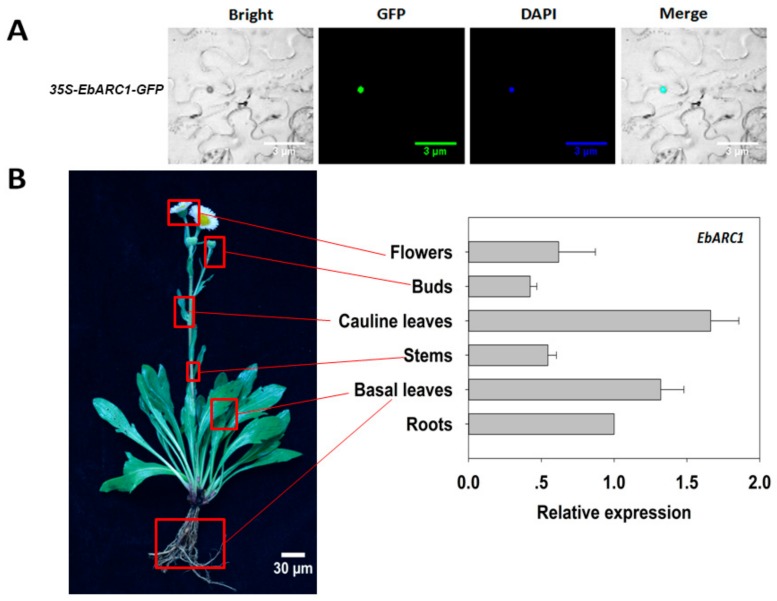
Expression analysis of *EbARC1*. (**A**) *N. benthamian* leaves were infected with 35S::*EbARC1*-GFP. Bright field, GFP, DAPI, and Merge images. (**B**) The tissue-specific expression patterns of *EbARC1* in flowers, buds, cauline leaves, stems, basal leaves, and roots of *E. breviscapus* determined by qRT-PCR analysis.

**Figure 4 ijms-21-01458-f004:**
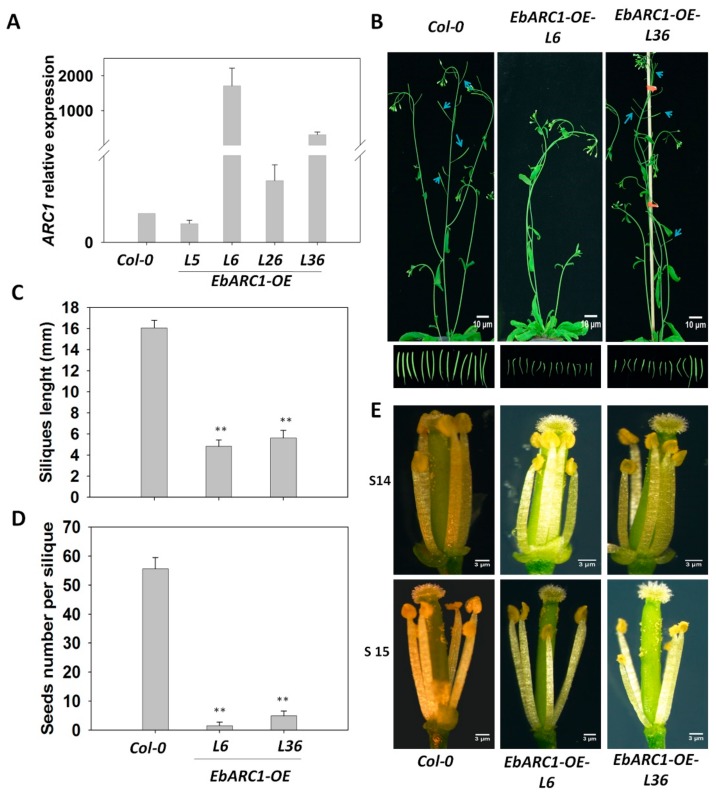
Expression of *EbARC1* resulted in infertility in *A. thaliana.* (**A**) qRT-PCR characterization of *EbARC*1 expression in four independent transgenic lines. (**B**) Phenotypes of *EbARC*1 overexpressed lines and wild type. (**C**) Silique lengths and (**D**) seed numbers of *EbARC*1 overexpressed lines and wild type. (**E**) Phenotypes of stamen and stigma in both overexpressed lines and wild type. Arrows indicate position of siliques. Error bars denote SDs (** *p* < 0.01).

**Figure 5 ijms-21-01458-f005:**
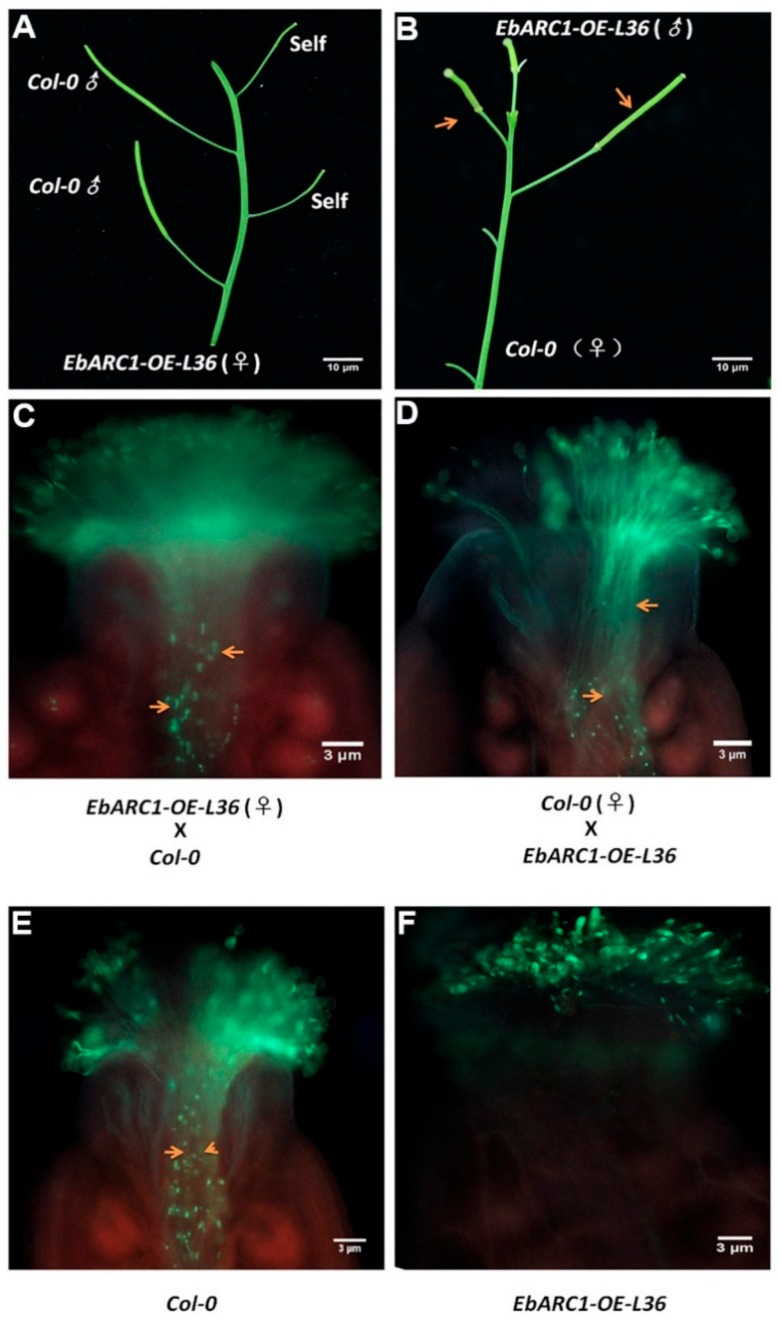
Overexpression of *EbARC1* conferred self-incompatibility (SI) in transgenic *A. thaliana*. (**A**) OEARC1-OE-L36 was (♀), while wild type and OEARC1-OE-L36 were (♂). (**B**) Wild type was (♀), while OEARC1-OE-L36 was (♂). (**C**) Pollen tube staining of OEARC1-OE-L36 (♀) and wild type (♂). (**D**) Pollen tube staining of wild type (♀) and OEARC1-OE-L36 (♂). (**E**) Pollen tube staining of selfing wild type. (**F**) Pollen tube staining of OEARC1-OE-L36. Crosses were carried out in three biological repeats (*n* = 20). Arrows indicate position of pollen tube.

**Figure 6 ijms-21-01458-f006:**
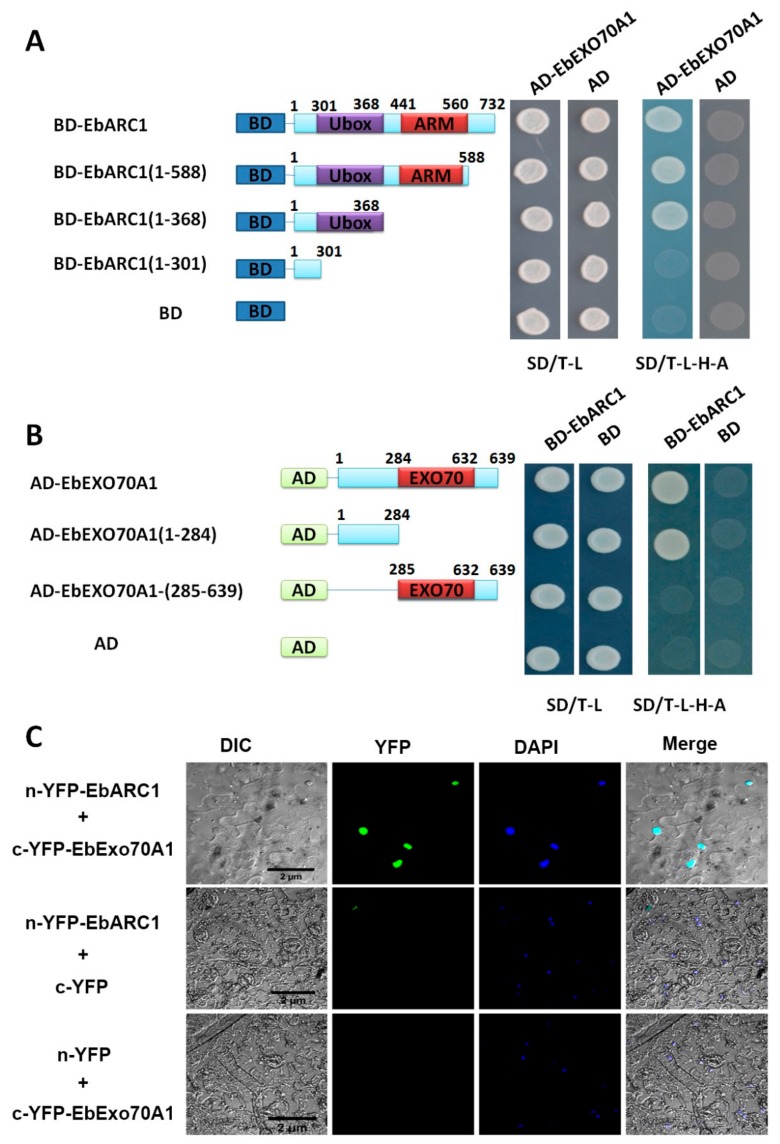
Protein interaction between EbARC1 and EbExo70A1. (**A**) Interaction between EbARC1 with EbExo70A1 was detected in the yeast two-hybrid (Y2H) system. pGBKT7 (BD) and pGADT7 (AD) were used as negative controls. The full-length EbARC1 and three truncated EbARC1 sequences were fused to the GAL4 DNA-binding domain (BD) in pGBKT7, while EbExo70A1 was inserted into the GAL4 activation domain (AD) in pGADT7. (**B**) The interaction between EbExo70A1 with EbARC1 were detected in Y2H system. The full-length EbExo70A1 and two truncated EbExo70A1 sequences were fused to the AD in pGADT7, while EbARC1 was inserted into BD in pGBKT7. (**C**) Interaction between EbARC1 with EbExo70A1 was detected in the bimolecular fluorescence complementation (BiFC) system. The N-terminal of YFP was fused to EbARC1 (n-YFP–EbARC1), while the C-terminal of YFP was fused to EbExo70A1 (c-YFP–EbExo70A1). DAPI (4’,6-diamidino-2-phenylindole) was used as a nucleus indicator.

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
