# Peer review of "EbARC1, an E3 Ubiquitin Ligase Gene in Erigeron breviscapus, Confers Self-Incompatibility in Transgenic Arabidopsis thaliana"

_ijms, 2020, doi:10.3390/ijms21041458_

Round 1
Reviewer 1 Report
The main aim of the paper is to study the mechanisms of genes that putative control the self-incompatibility of Erigeron breviscapus in Arabidopsis thaliana.
The information is very complete and can be useful about the SI of this species. However, more discussion is needed and the conclusions need to be expanded and more comparisons with other studies and results are needed. More information in the methodology is needed.
Specific comments
Title: Erigeron breviscapus not Erigeron Breviscapus. Not capital letter in the second word.
Line 39. Reference needed.
Line 52. All the name of a species in Latin should be followed by the name of the author the first time that it appears in the text. Eg. Brassica napus L. The same for the other Latin names of plants in the text.
Line 61. A more detailed description of the species is needed. Also about the fertilization system and the morphology of the flowers and reproductive system.
Line 74. Number of plants tested in each experiment? There were repetitions in different seasons or environments?
Line 111. How many plants were used? Repetitions in different environments?
Line 118. How many plants were used? Repetitions in different environments?
Line 149. How many crosses were made for each treatment?
Line 203. Transgenic lines L5 and L25 appear in the figure 4 but not in the text, why?
Line 304. Further discussion is needed. Which are the following steps that can be done? Mutations to EbARC1 can avoid SI in Arabidopsis? Can be mutated Erigeron or transformed to see if SI can be avoided? Practical applications? Avoiding SI can improve the yield of this species?
Author Response
Reviewer #1:
Comments and Suggestions for Authors
The main aim of the paper is to study the mechanisms of genes that putative control the self-incompatibility of Erigeron breviscapus in Arabidopsis thaliana. The information is very complete and can be useful about the SI of this species. However, more discussion is needed and the conclusions need to be expanded and more comparisons with other studies and results are needed. More information in the methodology is needed.
Response: We appreciate very much the overall comments. We have further improved the quality of the discussion, which highlighted with yellow color in the revised manuscript.
Specific comments
Title: Erigeron breviscapus not Erigeron Breviscapus. Not capital letter in the second word.
Response: Sorry, we have corrected this mistake.
Line 39. Reference needed.
Response: We have added references to the revised manuscript.
Line 52. All the name of a species in Latin should be followed by the name of the author the first time that it appears in the text. Eg. Brassica napus L. The same for the other Latin names of plants in the text.
Response: We have addressed these mistakes in the revised manuscript.
Line 61. A more detailed description of the species is needed. Also about the fertilization system and the morphology of the flowers and reproductive system.
Response: Thanks for your suggestion. Detailed content mentioned by you has been descripted in the “Introduction” section.
Line 74. Number of plants tested in each experiment? There were repetitions in different seasons or environments?
Response: Thank you for your question. In fact, we carried out three independent biological repeats, and the number of plants detected in each repeat was n = 30. The seedlings of E. breviscapus are all from the company base, and then transplanted to the natural greenhouse for normal growth. Three biological repeats are collected in the same environment with different seasons(Spring, summer and Autumn). Since the self incompatibility of E.breviscapus has been reported in previous article (Zhang, W.; Wei, X….. Yang, S.C. Transcriptomic comparison of the self-pollinated and cross-pollinated flowers of Erigeron breviscapus to analyze candidate self-incompatibility-associated genes. BMC Plant Biol. 2015, 15(1), 248-248), we did not consider doing it in different environments. Instead, we only further confirm whether this characteristic is consistent with our research purpose. Relevant contents have been described in “Materials and methods” and “Figure 1 legend”.
Line 111. How many plants were used? Repetitions in different environments?
Response: Thank you for your question. Crosses were carried out in three biological repeats (n=20). All the repetitions were only performed in an artificial greenhouse at 22°C under a 10 h light/14 h dark. Relevant contents have been revised in “Figure 5 legend”.
Line 118. How many plants were used? Repetitions in different environments?
Response: Thank you for your question. Ten E.breviscapus were mixed as a biological repeat. The PCR reactions were carried out in three biological and technical repeats. Relevant contents have been revised in “Materials and methods”.
Line 149. How many crosses were made for each treatment?
Response: 30 crosses were made for each treatment. Three biological repeats were carried out in spring, summer and autumn, respectively (n=30).
Line 203. Transgenic lines L5 and L25 appear in the figure 4 but not in the text, why?
Response: In Figure 4, the expression levels of L5 and L26 lines were not very high. Although we observed that the fertility of L5 and L26 lines was also weakened, it was not obvious compared to L6 and L36. Therefore, L6 and L36 lines were used for later experiments.
Line 304. Further discussion is needed. Which are the following steps that can be done? Mutations to EbARC1 can avoid SI in Arabidopsis? Can be mutated Erigeron or transformed to see if SI can be avoided? Practical applications? Avoiding SI can improve the yield of this species?
Response: In traditional cross breeding, inhibiting self-pollination by artificial emasculation or emasculation agent not only takes time, but also increases the breeding cost. In this study, the heterologous expression of EbARC1 in A. thaliana showed that EbARC1 functions in regulating SI response by transforming A. thaliana SC into SI. Considering that we have preliminarily established the Agrobacterium-mediated genetic transformation system of E. breviscapus, the site directed mutation of ARC1 in E. breviscapus through gene editing will help to verify the function of ARC1 and create homozygous parent materials in the future.

Reviewer 2 Report
Dear Editor,
the article “EbARC1, an E3 Ubiquitin Ligase Gene in Erigeron Breviscapus, Confers Self-incompatibility in Transgenic Arabidopsis” by Chen et al. is an interesting article dealing with the type of self incompatibilty of Erigeron breviscapus.
The genetic analysis of two genes of self incompatibility is toroughly carried out, also with 2 hybrd assay.
As a botanist i would prefer a couple of data more about the plant Erigeron breviscapus (author’s name, family). Name of genera and species shoud be in italics
The discussion should be probably improved
Line 43 to be either sporophyte SI (SSI) or gametophyte
Author Response
Reviewer #2:
Comments and Suggestions for Authors
Dear Editor, the article “EbARC1, an E3 Ubiquitin Ligase Gene in Erigeron Breviscapus, Confers Self-incompatibility in Transgenic Arabidopsis” by Chen et al. is an interesting article dealing with the type of self incompatibilty of Erigeron breviscapus. The genetic analysis of two genes of self incompatibility is toroughly carried out, also with 2 hybrd assay.
Response: We appreciate very much the overall comments.
As a botanist i would prefer a couple of data more about the plant Erigeron breviscapus (author’s name, family). Name of genera and species shoud be in italics
Response: We have addressed these mistakes in the revised manuscript.
The discussion should be probably improved
Response: Thanks for your suggestion. We have further improved the quality of the discussion, which highlighted with yellow color in the revised manuscript.
Line 43 to be either sporophyte SI (SSI) or gametophyte
Response: Thank you. The error has been corrected in the revised manuscript.
Round 2
Reviewer 1 Report
The main aim of the paper is to study the mechanisms of genes that putative control the self-incompatibility of Erigeron breviscapus in Arabidopsis thaliana.
Improvements have been made, all the changes have been made and the paper improved except
Line 84. Number of plants tested in each experiment? There were repetitions in different seasons or environments?
Line 124. How many plants were used? Repetitions in different environments?
In the author response some information was given but it is not reflected in the manuscript.
Author Response
Dear reviewer
We appreciate your suggestion. The describtion has been done in the revised manuscript. Hope to get your approval. Thanks a lots
Round 3
Reviewer 1 Report
The suggestions have been followed.